# OpenReview forum: "Unlearning or Obfuscating? Jogging the Memory of Unlearned LLMs via Benign Relearning"
_ICLR.cc/2025/Conference — ICLR 2025 Poster_

### Official Review · Reviewer_ah7J · 2024-10-27

**Soundness:** 3
**Presentation:** 3
**Contribution:** 4
**Rating:** 8
**Confidence:** 4

**Summary:**

The authors show that LLMs which have been subject to unlearning can still produce the learned information after additional finetuning on specific data (albeit data that does not contain the unlearned content), which the authors call "relearning".

**Strengths:**

- Figures 1 & 2 are nice schematics
- The experiments have a nice breadth and depth. I appreciate Section 6 for trying a simple experiment to identify the mechanism (even if the mechanism feels obvious)

Edit: I gave the authors clear instructions for what I thought would improve the paper:

> I would be happy to further revise my score to a 8 if (1) the title/abstract/main text is rewritten to more clearly highlight that these results demonstrate that current unlearning methods are simply obfuscating the information, not removing the information, from the networks, (2) the methodology is made more compact, (3) the current results polished and (4) additional supporting results added, e.g., visualizations of multiple metrics confirming consistency of metrics

I feel like the authors have done an admirable job improving the narrative of the paper. I added a few remaining minor suggestions that the authors can optionally integrate, but I'll be increasing my score to an 8.

**Weaknesses:**

- Section 3.1’s notation is fine until line 103 introduces “uninformative/unrelated” text. How does one quantify whether the model outputs uninformative/unrelated text? This feels like a very fuzzy notion and one that might be hotly contested in the research community.
- The utility of Table 1 is confusing to me. As I understand, this Table 1 is meant to be an example of how information can be relearned, but it doesn’t communicate any mechanism or intuition. Rather, it just states, “First, here is the original output. Then, here’s the output after unlearning. Finally, here’s the output after relearning.”
- Nit: Table 1: I personally would recommend adding column titles for TOFU and WHP.
- Table 1 Caption: After reading the section, I realized that Table 1 isn’t a schematic but is a real example? If so, the phrasing of the caption is ambiguous to me. I thought “Example of how keywords unseen … could still be recovered” meant “Here’s a demonstration of what unlearning->relearning would look like”, not “Here is actual data from our experiments”. Perhaps this could be made more clear.
- The same feedback for the Table 1 Caption applies to the Table 2 Caption
- Line 242-243: Citation(s) are needed for gradient ascent in “We unlearn the Phi-1.5 model (Li et al., 2023) using gradient ascent, a common unlearning baseline.” I agree that this is a common unlearning baseline, but pointers to previous work should be given.
- Nit: Line 251: “on the a set of text”
- Figure 3: I might have missed something, but which metric is being visualized as the “Attack Success Rate”? In 3.3, we were introduced to three metrics for the success/failure of unlearning and relearning, and I’m not sure which is visualized in Figure 3. It is thus difficult to know how to interpret these results.
- In Figure 3, since ASR is supported on [0, 1], please increase the colormap range from 0 to 0.8 to 0.0 to 1.0
- In Figure 3, please include Unlearning Steps = 0. I think this is important for understanding what the behavior of the model is before unlearning begins and better contextualizes the changes caused by unlearning and then relearning.
- In Figure 3, please add TOFU and WHP as axis titles. If you’re visualizing these heatmaps in matplotlib, I suggest axes[0].set_title(“TOFU”) and axes[1].set_title(“WHP”)
- I think that the data in Figure 3 might be better visualized as a lineplot: ASR on y, Relearning Steps on X and Unlearning Steps as the hue (including 0 unlearning steps!). I feel the heatmap makes understanding the results unnecessarily complicated.
- Section 4 is light on results. At a minimum, all three metrics should be plotted and included in the paper (either in the main text or appendix). The reader should know whether the results are consistent across the three metrics.
- Figure 4: Please use axis titles to label the models.
- Figure 4: I dislike when two adjacent figures using the same y axis have different y limits. It feels misleading to me.
- Figure 4: I recommend removing one of the two legends; there’s little point in duplicating that information.
- Structurally: I feel like the manuscript would be much more exciting if the results appeared faster than page 5. I felt like Section 2 might be moved to the end and Section 3 made much shorter (with a longer version perhaps deferred to the Appendix). Also Algorithm 1 has very little information but is quite verbose.
- Story: By the end of the paper, I feel like the biggest takeaway is that unlearning hasn’t actually removed the information from these networks but has instead made it superficially inaccessible such that a little bit of the right finetuning can recover that information. If anything, this paper feels like an indictment of the inadequacy of unlearning methods (or perhaps we should call them “obfuscating” methods rather than “unlearning” methods). It’s not clear to me that the title, abstract or story communicates this point. If the authors agree, I strongly urge them to change the title and rewrite the paper to make this clear.

I feel like there is a much more impactful story to be told here using these results, specifically that current unlearning methods are actually obfuscating methods and/or current unlearning methods do not cause the model to unlearn.

**I would be happy to revise my score to a 6 if (1) the methodology is made more compact and (2) the current results polished**

**I would be happy to further revise my score to a 8 if (1) the title/abstract/main text is rewritten to more clearly highlight that these results demonstrate that current unlearning methods are simply obfuscating the information, not removing the information, from the networks, (2) the methodology is made more compact, (3) the current results polished and (4) additional supporting results added, e.g., visualizations of multiple metrics confirming consistency of metrics**

**Questions:**

- On Line 421, how do you ensure that the unlearned model has 0% success rate in generating Anthony Mark?
- In Figure 5, should the legend be “Anthony | x” and “Mark | x Anthony”?
- Why are different metrics reported for different experiments in Sections 4 through 6? Can the same 3-4 metrics (the authors mention using 3 metrics, but then report NLL loss in Figure 5) not be plotted for every experiment?

Perhaps two related citations: https://arxiv.org/abs/2401.01814 and https://ai.stanford.edu/~kzliu/blog/unlearning (note, I'm unaffiliated with both)

---

> ### Author Response · Authors · 2024-11-22
> **Response to Reviewer ah7J**
>
> We appreciate the reviewer’s time and the valuable comments and feedback for our work. We address the concerns and questions below, and have made several updates to the paper itself based on your helpful suggestions.
>
> > **Section 3.1’s notation is fine until line 103 introduces “uninformative/unrelated” text. How does one quantify whether the model outputs uninformative/unrelated text? This feels like a very fuzzy notion and one that might be hotly contested in the research community.**
>
> By saying uninformative/unrelated, we mean the unlearned model should have low accuracy when evaluated on queries whose answers rely on knowledge of the unlearn set information. This is how existing empirical LLM unlearning benchmarks demonstrate “success” in unlearning. For example, if unlearning is successful, the knowledge memorization score for WMDP would be extremely low compared to the pre-unlearned model, and the keyword memorization accuracy for TOFU/WHP is close to 0%.
>
> > **Weaknesses 2-5 about Table 1, 2**
>
> Thanks for your suggestion. We have modified the captions to make it clearer that we are showing an actual example from our experiments.
>
> > **Weaknesses 6 about citation**
>
> Thanks for pointing this out. We have added a citation (Golatkar et al 2020) for gradient ascent.
>
> > **Weaknesses 7 about minor edits**
>
> Thanks for pointing this out. We have update the paper to fix this.
>
> > **Weaknesses 8-11 about Figure 3**
>
> Thank you, we have included the baseline where unlearning steps is 0. We also included a line plot of the same result in Appendix J for readers who prefer to see that visualization relative to the heatmap.
>
> > **Weaknesses 12 + Questions 3: Section 4 is light on results. At a minimum, all three metrics should be plotted and included in the paper (either in the main text or appendix).**
>
> Here we clarify that we use standard metrics/evaluation procedures for different unlearning tasks, as not all metrics are appropriate or informative for all tasks. For example, for our WMDP experiment where the goal is to recover unlearned knowledge, metrics like Rouge-L score would be a weak measure as sentences with different subsequences could have similar semantic meanings (see Table 2 as an example). Further, existence of a certain keyword also does not mean the answer is correct / relevant (see Appendix C.5.3—there isn’t an obvious set of keywords that can separate the original answer and relearned answer). We provide a detailed explanation and justification of the metrics/evaluation procedures used in our experiments in Appendix I; in general, these evaluation procedures match (and in some cases expand upon) the evaluations performed in the unlearning benchmarks we study.
>
> > **Weaknesses 13-15 about Figure 4**
>
> Thanks for the suggestion. We have modified the presentation of Figure 4 with aligned y-axis, deduplicated legends, and subplot titles.
>
> > **Questions 1: how do you ensure that the unlearned model has 0% success rate in generating Anthony Mark?**
>
> Similar to existing unlearning benchmarks, we measure this empirically: We use 100 random queries that start with a list of random English names and end with Anthony. If 0/100 model completions do not contain the pair Anthony Mark, we treat the model as successfully unlearning the information Anthony Mark. For every row in Table 5, we start with an unlearned model that has 0% accuracy in generating Anthony Mark pair for 100 random queries.
>
> > **Questions 2 about improved presentation**
>
> Thanks for the suggestion. We have modified the presentation of Figure 5 with the updated legend.
>
> > **Weaknesses 16-17 about structure and story**
>
> We greatly appreciate your suggestions to improve/clarify the story and the general exposition. We have made a few revisions to the paper in response:
> - We moved Algorithm 1 to the appendix.
> - We moved the related work section to the end of the paper.
> - We agree with the reviewer that an important take-away of the paper is that existing unlearning methods are not truly unlearning all information (it is just obfuscated/suppressed), which is what allows us to target the latent retained knowledge through relearning attacks. We also agree it would be beneficial to explain this more directly early on in the paper. We’ve updated the introduction and paper title to better reflect this point.

---

> > ### Author Response · Authors · 2024-11-25
> > **Follow up on rebuttal**
> >
> > Dear Reviewer,
> >
> > We are sorry to bother you, but since there is only one day left in the discussion period, we wanted to check in with you quickly to see if our rebuttal has addressed your concerns and have a chance to address any new ones.
> >
> > Thanks again!
> > Authors

---

> ### Comment · Reviewer_ah7J · 2024-11-26
> **Response to Authors' Round 1 Response**
>
> **Overall, I'm much happier with this paper and I feel very comfortable increasing my score to an 8.**
>
> Other suggestions:
>
> **(Minor) Title:** The new title is an improvement, but I feel it could still be improved further. Perhaps something like "Unlearning or Obfuscating? Undoing Unlearning via Benign Relearning" or "Unlearning or Suppressing? Undoing Unlearning via Benign Relearning". I won't take this into account but I urge the authors to continue brainstorming :)
>
> Other possible suggestions:
> - Machine Unlearning's False Promise: Recovering 'Forgotten' Information via Benign Relearning
> - The Persistence of Knowledge: On the Limitations of Current LLM Unlearning Methods
> - The Illusion of Unlearning: Recovering 'Forgotten' Knowledge in LLMs via Benign Relearning
>
> Suppression was another word the authors used that I liked.
>
> **(Minor) Abstract:** I think the abstract could be slightly improved by concluding with a sentiment like: "The paper's core contribution seems to be revealing that current unlearning approaches are fundamentally flawed because they don't truly remove information from the model. Instead, they appear to be creating a superficial barrier that can be circumvented with relatively simple techniques. This challenges the very concept of "unlearning" as it's currently implemented." As currently written, the abstract feels a little too narrow and could benefit by adding broader context for why this work is significant.
>
> **(Minor) Citation Formatting:** At the bottom of page 1, the citation box wraps to the header of Page 2. This is minor but hopefully can be fixed.
>
> **(Minor) Related Concurrent Work:** I realize now that the relearning studied in this paper is somewhat akin to the parameter-perturbation memorization analysis conducted by https://arxiv.org/abs/2406.14549. Perhaps worth citing? It is likely concurrent work. You might also want to reach out to the authors about citing your paper. The conclusions are spiritually similar, although the settings and methods are quite different.

---

> > ### Author Response · Authors · 2024-11-27
> > **Response**
> >
> > We are glad that our response addresses some of your concerns and thank you very much for your suggestion and updating the score! We have updated the paper with the following changes:
> > - We added a concluding sentence in the abstract highlighting a broad takeaway from this study.
> > - We fixed the citation formatting.
> > - Thanks for pointing out this work. We believe this is related to the construction we have for the synthetic data experiment. We provide a citation for this work in Section 6.

---

### Official Review · Reviewer_k5DU · 2024-10-31

**Soundness:** 3
**Presentation:** 4
**Contribution:** 2
**Rating:** 5
**Confidence:** 2

**Summary:**

The paper discusses the vulnerability of machine unlearning processes in LLMs to targeted relearning attacks. Machine unlearning is designed to remove sensitive or unwanted data from training datasets to comply with regulations like GDPR or to enhance privacy. However, the authors demonstrate that with minimal and loosely related data, it is possible to reverse the effects of unlearning, effectively "jogging" the memory of LLMs to recall or reproduce the unlearned information. The authors conducted experiments on three representative LLM unlearning benchmarks, aiming to solidify the observations of their proposal.

**Strengths:**

LLM unlearning is an important topics that has received increasing attention these days, while we still lack a detailed understanding of the key mechanism behind it. The exploration of the relearning can contribute to the literature in the sense that unlearning might not be that easy as we previously think, as the recovering of unlearned knowledge indicates that they are still parameterised in the models after unlearning. It may also indicate that the current evaluation metrics of unlearning might be insufficient, as they typically suggest that many methods, such as GA, can fully remove the unlearned knowledge (though the retain performance is also affected).

The authors conducted experiments across a set of different unlearning benchmarks, backbones, methods, and setups. All the results align with the authors' observations that relearning can occur after unlearning. Also, the motivation figures are good.

**Weaknesses:**

My main concern is about the novelty, as many of the previous works have already suggested relearning can occur after unlearning. As suggested by the authors, the only difference over previous works is that this paper separates the data for relearning in a more solid and rigorous way. Also, the authors try to explain why relearning can occur, but the explanation, in my personal point of view, is a little bit of trivial. All the experiments the authors conducted are already under the assumption that the data for relearning and the data for unlearning has some strong correlation, so the explanation in Section 6 does not provide further information, considering the main explanation of "When the unlearned set contains highly correlated pairs of data, relearning on only one can more effectively
recover information about the other." I am also quite confused about the claim of "while these heuristics successfully limit LLMs’
power to generate samples xfrom Du, they do not remove associations between different parts of x." Since we minimise the conditional probability of p(x_i|x_<i), seemingly all the correlation within x will be destroyed. Please forgive me if I made any misunderstanding and kindly please explain more about this point. BTW, I think a possible way to improve the contribution of this paper is to explore new methods that are robust to relearning attacks, mainly considering the fact that the scenario of relearning is somewhat well-knowned in the literature.

I appreciate the authors to follow the classical definition of unleanring in eliminating the influence of data targeted to be unlearned. However, as suggested by [1], there are another possible goals of unlearning in the literature of unlearning: to much extent eliminate the knowledge targeted to be unlearned. I am not quite sure about it, but it seems that the "to much extent" goal may not more suitable for LLM unlearning since we want the possibility in generating unlearned data to be exactly zero.

[1] Rethinking Machine Unlearning for Large Language Models

**Questions:**

Kindly please refer to the section of weaknesses.

---

> ### Author Response · Authors · 2024-11-22
> **Response to Reviewer k5DU 1/2**
>
> We appreciate the reviewer’s time and the valuable comments and feedback for our work. We address the concerns and questions below:
>
> > **My main concern is about the novelty**
>
> We summarize the novelty of our work below:
> - As highlighted in the general response, a major difference between our work and prior work in relearning is that we carefully study the scenario where the relearn set itself is insufficient to infer knowledge in the evaluation set. This ensures that the attack is actually “relearning” instead of simply “learning” the knowledge. We believe this is an important distinction to make and has not been rigorously studied in prior work. As a result, we are able to more accurately claim that existing unlearning methods do not actually forget the forget set data, and our work shows that relearning attacks may be much more prevalent/practical than what was known from prior work. We have modified the intro and added an additional Appendix G to better emphasize these points.
> - Additionally, our work differs by providing simple explanations about why and when relearning works. This includes our Section 5 on synthetic data experiment, as well as our new experiment on the relation between relatedness of relearn text and the quality of relearning (see our general response for more details).
> - Based on our empirical results, we provided practical guidelines for LLM unlearning. See our general response for more details.
>
> > **All the experiments the authors conducted are already under the assumption that the data for relearning and the data for unlearning has some strong correlation, so the explanation in Section 6 does not provide further information**
>
> The main goal of the previous Section 6 (current Section 5) is to investigate why and when relearning will happen in a simplified setting where we can manually control the relevance between a pair of data (in this experiment, the relation between Anthony and Mark). Experiments on this synthetic dataset shows that **the quality of relearning depends on the correlation strength between the relearn set $D'$ and the target knowledge**. However, while correlation helps to explain why relearning can be effective, we do not find that “strong” correlation is necessarily needed for relearning to be successful. Indeed, our non-synthetic experiments show that benign, loosely related data such as wiki data or general knowledge of similar keywords can be used for relearning, which makes such an attack practical and possibly quite hard to defend against given that most harmful or private knowledge also shares at least some relation with benign/public knowledge.
>
> Upon reflecting on the reviewer’s comments, we have run additional experiments on the synthetic data and WMDP benchmark to explore this idea in more detail—systematically exploring the effectiveness of the relearn text with different levels of relatedness to the forget set, beyond the simplified “correlation” experiments. The results are shown in the general response. These results highlight that while finetuning on irrelevant text or text from the retain set can provide a marginal boost to the attack success rate, relearning is more effective if there is a stronger degree of relatedness of the relearn text.
>
> > **Since we minimise the conditional probability of p(x_i|x_<i), seemingly all the correlation within x will be destroyed.**
>
> While the process of unlearning ensures that the model does not output $x_i | x_{<i}$, our work shows that these correlations may still persist in the unlearned model. In particular, while it’s true that the unlearning objective aims to optimize $L_i = log p_w(x_i|x_{<i})$ for all $i$, it does NOT mean these log probabilities are independent when $w=w_U$, the unlearned model. Specifically, given a highly correlated pair of tokens $(x_k,x_{k+1})$, $L_{k+1}$ changes as $L_k$ changes, as shown in Figure 5. This exactly provides evidence that approximate unlearning heuristics are not robust, as we see that the two probabilities are not independent.
>
> > **BTW, I think a possible way to improve the contribution of this paper is to explore new methods that are robust to relearning attacks, mainly considering the fact that the scenario of relearning is somewhat well-knowned in the literature.**
>
> Thanks for the suggestion. In this work our main focus is to study the extent to which current unlearning heuristics for LLMs are vulnerable to relearning. We believe coming up with a fully comprehensive defense against relearning may be quite difficult, and would be an interesting direction of future (significant) study. However, we do point out several guidelines that we believe can help to inform such defense methods, including highlighting metrics, methods, and models from our study that seem more promising in terms of preventing against relearning (Section 6).

---

> ### Author Response · Authors · 2024-11-22
> **Response to Reviewer k5DU 2/2**
>
> >  **I appreciate the authors to follow the classical definition of unleanring in eliminating the influence of data targeted to be unlearned. However, as suggested by [1], there are another possible goals of unlearning in the literature of unlearning: to much extent eliminate the knowledge targeted to be unlearned. I am not quite sure about it, but it seems that the "to much extent" goal may not more suitable for LLM unlearning since we want the possibility in generating unlearned data to be exactly zero.**
>
> In this work we consider multiple possible unlearning goals. In fact, our experiments cover three popular unlearning goals which themselves are quite different: unlearning verbatim memorized text, unlearning memorized knowledge, and unlearning memorized keywords (in the paper you sent, the first relates to what the authors refer to as ‘data influence removal’, whereas the other two relate to ‘model capability removal’). We find that regardless of the underlying goal of unlearning, all scenarios are susceptible to relearning attacks. Please let us know if we have misunderstood your comment.

---

> ### Comment · Reviewer_k5DU · 2024-11-24
>
> Many thanks for the responses from the authors. I sincerely appreciate your efforts that address some of my confusion, but, just in my opinion, the contribution of this paper is not sufficient enough. I would like to raise my scores to 5 but decrease my confidence to 2, I cannot further raise my score, sorry! If the paper is not accepted, I suggest the authors could delve deeper into the reasons behind relearning with some more formal verifications. Also, I think the authors should explore methodologies to overcome relearning. Good luck!

---

> > ### Author Response · Authors · 2024-11-25
> >
> > We are glad that our response addresses some of your concerns and thank you very much for updating the score!

---

### Official Review · Reviewer_FNSp · 2024-11-02

**Soundness:** 3
**Presentation:** 3
**Contribution:** 2
**Rating:** 8
**Confidence:** 4

**Summary:**

This paper explores 'relearning' attacks on language models that have had a part of their dataset removed/suppressed using some machine unlearning technique. They show that in their examples training the 'unlearned' models on a related dataset/portion of the unlearning dataset causes performance on the unlearning dataset/holdout portion of the unlearning dataset to improve, and attempt to characterise this (in part) by considering correlations in the datasets.

**Strengths:**

- overall well-written and easy-to-follow paper.
  - some formatting (?) errors which obscure some maybe-important information (e.g. line 424, "Next, we construct the forget set D_u by collecting all the data that on a subset of the names." seems to be truncated)
- very clear discussion of limitations in previous works owing to overlap between relearn and evaluation data, and a useful factorisation into retraining on unlearning data vs retraining on related data.
- good/comprehensive selection of unlearning methods and datasets to evaluate.
- the experiment in section 6 is especially interesting and similar work to this would be a promising avenue for better understanding the situations in which relearning is a danger/possibility.
- useful evidence is gathered for demonstrating that LoRA unlearning is less effective against relearning than full-parameter fine-tuning.

**Weaknesses:**

- not especially novel; some previous works (e.g. Lynch et al 2024, among others) have explored the relearning setting, and some have considered "the relation between relearn set and the queries used for evaluation" (e.g. Guo et al 2024 - "Robust Unlearning via Mechanistic Localization"). nonetheless it is still productive to have more work studying this.
    - exploring the themes touched on in section 6 in greater detail/with more [or more complex] examples (e.g. using a setting similar to TOFU, perhaps introducing artificial correlations/decorrelations between the unlearning/relearning datasets), or attempting to find a quantative relationship between e.g. correlation and relearning, would have strengthened the evidence for takeaway #3, which otherwise had relatively weaker evidence than takeaways #1 and #2.

**Questions:**

- did you manage to find any heuristics/quantative rules for predicting when relearning is especially easy/hard? does e.g. model scale, ROUGE or embedding-based similarity metrics, number of unlearning steps predict when relearning is easy/hard?

---

> ### Author Response · Authors · 2024-11-22
> **Response to Reviewer FNSp**
>
> We appreciate the reviewer’s time and the valuable comments and feedback for our work. We address the concerns and questions below:
>
> > **some formatting (?) errors which obscure some maybe-important information**
>
> Thank you for pointing this out! We have made the edits to our revision.
>
> > **not especially novel; some previous works (e.g. Lynch et al 2024, among others) have explored the relearning setting**
>
> A key contribution of this work is that we evaluate the unlearning-relearning pipeline with careful construction of relearn set and eval set, which has not been explored in previous work (Tamirisa et al. (2024/3), Lynch et al. (2024/2), Qi et al. (2023/10)). We believe the composition of the relearning set (and its relation to evaluation) is critical to study, as the fact that relearning can be done even when the data does not provide direct answers to the eval set is quite surprising, and shows that unlearned models may be quite susceptible to a wide variety of practical attacks. Please refer to our general response for a more detailed discussion on differences with prior works and contributions.
>
> > **some have considered "the relation between relearn set and the queries used for evaluation" (e.g. Guo et al 2024 - "Robust Unlearning via Mechanistic Localization").**
>
> We appreciate the reviewer for pointing out this work, which we agree explores a similar setting as our partial unlearn set attack. However, please note that this work was submitted to arxiv on 10/16/2024 (after the ICLR submission deadline), and we were thus unaware of the work at the time of submission. Unlike this work, our work also explores a number of relearning attacks using public information, and we provide simple explanations of when/why relearning will work through synthetic data experiments.
>
> > **perhaps introducing artificial correlations/decorrelations between the unlearning/relearning datasets), or attempting to find a quantative relationship between e.g. correlation and relearning, would have strengthened the evidence for takeaway #3**
>
> Thank you for your suggestion. To better understand how different relearn data can recover forget set information from LLMs, we’ve run additional experiments on the synthetic data by choosing relearn text with different levels of relatedness to the forget set. The results are shown in the general response. These results highlight that while finetuning on irrelevant text or text from the retain set can provide a marginal boost to the attack success rate, relearning is more effective if there is a stronger degree of relatedness of the relearn text.
>
> > **did you manage to find any heuristics/quantative rules for predicting when relearning is especially easy/hard?**
>
> As we summarized in Section 6 (discussion), we did identify a number of heuristics/rules for predicting when relearning is easy/hard.
> - Unlearning task and method: As highlighted in our guidelines in Section 6 and Table 6 in Appendix E, relearning difficulty depends on what unlearning task one evaluates on and what method one uses for unlearning. For example, compared to other unlearning heuristics, RMU seems to be a strong method to protect against relearning when the goal of unlearning is to limit the model’s power to generate high quality text given a query. On the other hand, RMU is very vulnerable to vanilla relearning when we provide multiple choices to the queries and the unlearning goal is to perform badly on multiple choice questions.
> - Unlearning steps: Figure 3 shows that more unlearning steps will lead to a deeper unlearned checkpoint, which tends to make relearning harder.
>
> Therefore, difficulty of relearning can vary based on the unlearned model itself, the unlearning goal, the relearning data at hand, and potentially other factors. We believe these would be promising directions to explore in future work in order to come up with more comprehensive defenses.

---

> > ### Comment · Reviewer_FNSp · 2024-11-25
> >
> > Thank you for your extensive reply and corresponding updates.
> >
> > > Thank you for pointing this out! We have made the edits to our revision.
> >
> > Thank you!
> >
> > > A key contribution of this work is that we evaluate the unlearning-relearning pipeline with careful construction of relearn set and eval set...
> > > We appreciate the reviewer for pointing out this work, which we agree explores a similar setting as our partial unlearn set attack...
> >
> > Thank you for the discussion of previous related work. My apologies for not realizing that the referenced paper was submitted to ArXiv after the ICLR submission date. Accordingly, I have updated my 'contribution' score to a 3, and my rating to an 8.
> >
> > > Thank you for your suggestion. To better understand how different relearn data can recover forget set information...
> >
> > Thank you for these results.

---

> > > ### Author Response · Authors · 2024-11-25
> > >
> > > We are glad that our response addresses your concerns and thank you very much for updating the score!

---

### Official Review · Reviewer_eNzV · 2024-11-03

**Soundness:** 4
**Presentation:** 4
**Contribution:** 3
**Rating:** 6
**Confidence:** 3

**Summary:**

The paper studies targeted relearning attacks on large language models (LLMs) that have undergone approximate unlearning of certain information. The key idea is that by relearning on a small amount of auxiliary data that is loosely related to the unlearned information, it is possible to "jog" the memory of the unlearned model and cause it to output the supposedly forgotten information. They test relearning attacks on three common unlearning applications: 1) removing harmful/hazardous knowledge, 2) mitigating memorization of copyrighted content, and 3) suppressing retention of specific undesirable words/phrases. In each case, they show it is possible to recover the unlearned information by relearning either on a limited subset of the unlearned data that doesn't contain the target information directly, or by relearning on publicly available information related to the unlearned data. They provide a simplified theoretical example showing that when unlearned data contains correlated pairs of tokens, relearning on one token can trigger recovery of the other.

**Strengths:**

1. Demonstrates an important limitation in current LLM unlearning methods, showing they don't fully remove knowledge
2. Tests the relearning attack on multiple datasets, unlearning applications, and base models
3. Provides both empirical results and theoretical intuition for why relearning can recover supposedly forgotten information
4. Presentation is clear and results are well summarised at the end.

**Weaknesses:**

The simplified example is a very interesting experiment, and I think a promising direction for better understanding of which finetuning data can recover harmful capabilities from LLMs. The paper would have benefited from more experiments here with different datasets, models and n gram correlations to study.

Some discussion of defenses that model providers like OpenAI could deploy on their finetuning apis to mitigate the threat posed here, such as keyword filtering, would be helpful for the research community.

The result here is useful but not too far away from what has been suggested by Qi et al. (https://arxiv.org/abs/2310.03693). As such I think the strength of the contribution is relatively weak.

**Questions:**

I have no questions

---

> ### Author Response · Authors · 2024-11-22
> **Response to Reviewer eNzV**
>
> We appreciate the reviewer’s time and the valuable comments and feedback for our work. We address the concerns and questions below:
> > **I think a promising direction for better understanding of which finetuning data can recover harmful capabilities from LLMs.**
>
> In this work we looked at three common unlearning benchmarks: WMDP, TOFU, and WHP and a number of common base models: zephyr-7b-beta, Llama-2-7b, and Llama-3-8b for different unlearning tasks. As per the reviewer’s suggestion, to further understand how different relearning data can recover forget set information from LLMs, we’ve run additional experiments on the synthetic data and WMDP benchmark by exploring relearning text with different levels of relatedness to the forget set. The results are shown in the general response. These results highlight that while finetuning on irrelevant text or text from the retain set can provide a marginal boost to the attack success rate, relearning is more effective if there is a stronger degree of relatedness of the relearn text.
>
> > **Some discussion of defenses that model providers like OpenAI could deploy on their finetuning apis to mitigate the threat posed here, such as keyword filtering, would be helpful for the research community.**
>
> Thank you for your suggestion. We agree that protection at input / output level would be interesting to look at. Our relearning attack does not conflict with attacks such as jailbreaking attacks whose goal is to break input / output space alignment techniques such as guardrails. Hence, it would be an interesting next step to try to combine relearning attacks with existing prompt-based attack methods to attack aligned models at multiple places within the LLM inference pipeline. We have added this as a direction of future work in the conclusion section.
>
> > **The result here is useful but not too far away from what has been suggested by Qi et al.**
>
> There are several key differences between our work and Qi et al.
> - Different scenario: In this work we explore finetuning-based relearning attacks to demonstrate vulnerabilities in unlearning, whereas Qi et al. uses finetuning to compromise safety training.
> - Focus on relearning data: Beyond the relearning attack itself, as we discuss in our Figure 2 and Section 2.2, 2.3, a key contribution of this work is that we characterize types of relearning data that may lead to successful attacks. Indeed, Qi et al points out (Remark 4) that understanding what types of data may lead to the most significant safety deterioration is a “critical future direction”. Our work helps to answer this question in the context of relearning: We formalize the notion of benign data (relearning data that cannot provide direct answers to the eval queries), and systematically explore when such benign data may lead to successful relearning attacks across common unlearning applications.
> - Explanation: Finally, our work also differs by helping to provide plausible explanations (Section 5) on why such a phenomenon may happen, including our analysis on the simplified keyword search example.
>
> Please also refer to our general response for more detailed discussion on contributions of this work.

---

> > ### Author Response · Authors · 2024-11-25
> > **Follow up on rebuttal**
> >
> > Dear Reviewer,
> >
> > We are sorry to bother you, but since there is only one day left in the discussion period, we wanted to check in with you quickly to see if our rebuttal has addressed your concerns and have a chance to address any new ones.
> >
> > Thanks again!
> > Authors

---

### Meta-Review · Area_Chair_wRtW · 2024-12-18

**Metareview:**

The paper studies targeted relearning attacks on large language models (LLMs) that have undergone approximate unlearning of certain information. The key idea is that by relearning on a small amount of auxiliary data that is loosely related to the unlearned information, it is possible to "jog" the memory of the unlearned model and cause it to output the supposedly forgotten information. They test relearning attacks on three common unlearning applications: 1) removing harmful/hazardous knowledge, 2) mitigating memorization of copyrighted content, and 3) suppressing retention of specific undesirable words/phrases. In each case, they show it is possible to recover the unlearned information by relearning either on a limited subset of the unlearned data that doesn't contain the target information directly, or by relearning on publicly available information related to the unlearned data. They provide a simplified theoretical example showing that when unlearned data contains correlated pairs of tokens, relearning on one token can trigger recovery of the other. The idea is interesting. The experiments are extensive and well-documented. For example, the useful evidence is gathered for demonstrating that LoRA unlearning is less effective against relearning than full-parameter fine-tuning. More importantly, this paper has a very clear discussion of limitations in previous works owing to overlap between relearn and evaluation data, and a useful factorisation into retraining on unlearning data vs retraining on related data. While the reviewers had some concerns about the novelty, the authors did a particularly good job in their rebuttal. Therefore, most of us have agreed to accept this paper for publication! Please include the additional discussion in the next version.

**Additional Comments On Reviewer Discussion:**

Some reviewers raise the score after the rebuttal.

---

### Decision · Program_Chairs · 2025-01-22

Accept (Poster)